# Evolution and Function of MADS-Box Transcription Factors in Plants

**DOI:** 10.3390/ijms252413278

**Published:** 2024-12-11

**Authors:** Zihao Zhang, Wenhui Zou, Peixia Lin, Zixun Wang, Ye Chen, Xiaodong Yang, Wanying Zhao, Yuanyuan Zhang, Dongjiao Wang, Youxiong Que, Qibin Wu

**Affiliations:** 1National Key Laboratory for Tropical Crop Breeding, Institute of Tropical Bioscience and Biotechnology/Sanya Research Institute, Chinese Academy of Tropical Agricultural Sciences, Sanya 572024, China; 15897575057@163.com (Z.Z.); zwh19961546644@163.com (W.Z.); linpx579@163.com (P.L.); qq742640354@gmail.com (Z.W.); yangxd426@163.com (X.Y.); zhaowanying@itbb.org.cn (W.Z.); zhangyuanyuan@itbb.org.cn (Y.Z.); dongjiaow@126.com (D.W.); 2Key Laboratory of Sugarcane Biology and Genetic Breeding, Ministry of Agriculture and Rural Affairs, National Engineering Research Center for Sugarcane, College of Agriculture, Fujian Agriculture and Forestry University, Fuzhou 350002, China; 3College of Landscape Architecture and Art, Fujian Agriculture and Forestry University, Fuzhou 350002, China; 2191775001@fafu.edu.cn

**Keywords:** MADS-box, evolution and function, stress response, crop improvement

## Abstract

The MADS-box transcription factor (TF) gene family is pivotal in various aspects of plant biology, particularly in growth, development, and environmental adaptation. It comprises Type I and Type II categories, with the MIKC-type subgroups playing a crucial role in regulating genes essential for both the vegetative and reproductive stages of plant life. Notably, MADS-box proteins can influence processes such as flowering, fruit ripening, and stress tolerance. Here, we provide a comprehensive overview of the structural features, evolutionary lineage, multifaceted functions, and the role of MADS-box TFs in responding to biotic and abiotic stresses. We particularly emphasize their implications for crop enhancement, especially in light of recent advances in understanding the impact on sugarcane (*Saccharum* spp.), a vital tropical crop. By consolidating cutting-edge findings, we highlight potential avenues for expanding our knowledge base and enhancing the genetic traits of sugarcane through functional genomics and advanced breeding techniques. This review underscores the significance of MADS-box TFs in achieving improved yields and stress resilience in agricultural contexts, positioning them as promising targets for future research in crop science.

## 1. Introduction

The MADS-box transcription factor (TF) gene family constitutes a significant subset within the broader category of transcription factor families. Eukaryotic MADS-box TF genes can be classified into two types, Type I and Type II. The former is characterized by the presence of the conserved MADS-box domain, whereas the latter possesses both the conserved MADS-box and K domains [1]. MADS-box genes are ubiquitous across animals, plants, and fungi, fulfilling essential biotic functions [2]. Notable examples include the serum response factor (SRF), which regulates the transcription of the proto-oncogene c-fos [3], and MCM1, which governs the transcription of cell type-specific genes that respond to pheromones in budding yeast [4]. In plants, MADS-box proteins serve as essential regulators for both the vegetative and reproductive growth. Extensive investigations have been conducted in several plant species such as *Arabidopsis thaliana*, rice (*Oryza sativa*), maize (*Zea mays*), and tobacco (*Nicotiana tabacum*). The modification of MADS-box genes involved in flower organ development can significantly alter traits such as petal number, color, and flowering time, making them particularly valuable in ornamental plant breeding [5]. For instance, overexpression of *AGL24* has been shown to promote flowering and facilitate the conversion of flower meristems into inflorescence meristems. Moreover, there is an interaction between *SOC1* and *AGL24* in the regulatory network governing flowering time [6,7]. Additionally, *AGL6* acts as a transcriptional suppressor, promoting the transition to flowering by directly inhibiting *ELF3* [6,7]. In tomatoes (*Solanum lycopersicum*), *FUL1* and *FUL2* exhibit redundant functions and are implicated in various aspects of the fruit ripening process, including cell wall modification, the production of cuticle components and volatile substances, as well as the accumulation of glutamic acid [8]. It is thus deduced that the regulation of MADS-box genes can influence the size, shape, flavor, and other characteristics of the fruit, thereby demonstrating their utility in fruit cultivation [8]. Furthermore, MADS-box TFs are associated with plant responses to stress, and manipulating these genes has the potential to develop crop varieties that exhibit enhanced stress resistance [9,10]. Silencing of *CaAGL8* significantly reduces both heat resistance and the sensitivity of pepper (*Capsicum annuum*) to bacterial infections under conditions of room temperature, as well as elevated temperature and humidity. In transgenic *Arabidopsis*, the overexpression of *DgMADS114* and *DgMADS115* has been shown to enhance the tolerance to polyethylene glycol (PEG), sodium chloride (NaCl), abscisic acid (ABA), and heat stress [11,12,13]. Regarding the involvement of MADS-box TF genes in various stages of plant growth and development, the expression modifications to these genes hold significant promise for enhancing biomass and yield, which could be crucial for plant breeding. For example, *AGL23* plays a role in the early phases of gametogenesis, *AGL80* influences central cell differentiation, and *AGL62* inhibits cell differentiation while promoting nuclear proliferation during the initial stages of endosperm development [14]. Interestingly, *NlMADS4*, *NlMADS30*, and *NlMADS69* are specifically and highly expressed during the development of rambutan (*Nephelium lappaceum*) peel, indicating their key roles in this process [15].

Previous reviews have compiled studies on the engagement of specific MADS-box TF genes in abiotic stress responses across various plant species [16]. However, due to the expanding breadth of research, more recent investigations into MADS-box gene responses under abiotic stresses, as well as their roles in biotic stress conditions—particularly within tropical crops—remain relatively sparse. Therefore, we construct phylogenetic trees using single-copy genes from the China National Knowledge Infrastructure (CNKI) and Web of Science (WOS) databases, encompassing primarily two decades of relevant literature. By integrating qualitative descriptions with quantitative analysis, this review collates information on the structure, classification, origin, evolution, and the latest insights into the involvement of MADS-box TFs in diverse stress regulation networks. Furthermore, it critically evaluates their prospective applications and potential impacts specifically within sugarcane (*Saccharum* spp.), a pivotal tropical crop [17,18,19]. This not only consolidates existing knowledge but also highlights emerging trends and gaps, fostering a deeper understanding of the versatile functions of MADS-box genes in stress resilience and adaptation strategies across plant species, with a focused exploration of sugarcane. The present study aims to bridge theoretical advancements with practical agricultural challenges, offering valuable insights for researchers and practitioners engaged in enhancing crop robustness against environmental adversities.

## 2. Structure and Classification of MADS-Box

### 2.1. Structure and Classification

MADS-box proteins typically consist of a conserved MADS-box domain, along with K-, C-, and I-domains (Figure 1a) [20]. The MADS-box domain, named after the four originally identified members, MCM1, *AGAMOUS* (*AG*), *DEFICIENS* (*DEF*), and SRF [21,22], comprises 56 amino acids and contains one α-helix and two β-sheets (Figure 1b,c). The DNA binding site, located at the N-terminal end, is critical for its function as a transcription factor. MADS-box proteins primarily target (CC(A/T)GG) sequences and similar motifs, collectively referred to as *CArG-box*. Binding to these elements within promoter regions enables MADS-box proteins to directly influence gene expression [23]. Interestingly, many defense genes, as well as floral organ-specific genes, contain *CArG-box* sequences in their promoter regions, including *PR1*, *PR2*, and *PR5*. Additionally, the promoter regions of related protein genes and floral organ-specific genes, such as *AGAMOUS*, *APETALA1*, and *LEAFY*, often include *CArG-box* sequences [24,25]. The I-domain, which follows the MADS domain, can form an α-helix that interacts with the α-helix in the MADS domain, thereby participating in dimer formation [26]. The K-domain, composed of a hydrophobic α-helix (Figure 1c), plays a significant role in the dimerization and oligomerization of MADS-box proteins, thus influencing their transcriptional regulatory functions [27]. Furthermore, it is crucial for facilitating interactions between MADS-box proteins and other transcriptional regulators, forming complexes that finely tune the expression of downstream target genes [28]. In *Arabidopsis*, the MADS-box protein SEPALLATA3 can cooperate with the *LEAFY* to regulate the expression of genes associated with floral organ development [29]. Similarly, the rice MADS-box protein *OsMADS14* can coordinate with the WUSCHEL-related homeobox protein *OsWOX13* to regulate the expression of flowering-related genes [30]. Additionally, the citrus (*Citrus reticulata*) MADS-box protein *CiAGL9* interacts with CiMADS43 and plays a role in citrus flowering and leaf development [31]. These regulatory processes are inextricably linked to the I- and K-domain, while the C-domain exerts substantial influence by mediating interactions between MADS-box proteins and other TFs or co-regulators, thereby affecting the modulation of transcriptional activity [32]. Phosphorylation of the AG protein can influence its capacity to bind to downstream genes, subsequently impacting the development of floral organs [33]. In kiwifruit (*Actinidia chinensis*), the expression of the MADS-box gene *AcFLCL* is regulated by histone modifications, particularly H3K4me3. This modification is associated with gene activation and transcriptional activity, suggesting that *AcFLCL* may involve epigenetic regulatory mechanisms in response to cold signals and the regulation of dormancy states [34]. In *Phalaenopsis amabilis*, the acetylation levels of H3K9K14ac in the translation initiation region (ATG) of the *PeMADS4* gene are significantly higher in the labium than in the petals, with a ratio that is 4.9 times greater. This finding indicates that elevated levels of H3K9K14ac may enhance the expression of the *PeMADS4* gene in the labium, thereby playing a crucial role in the morphogenesis of phalaenopsis [35]. These epigenetic regulatory mechanisms are closely linked to the C-domain.

The MADS-box TFs are extensive, comprising numerous members that can be categorized into subclasses based on various criteria. According to the sequence homology of their MADS-box domains, they are classified into two major types, Type I and Type II. Type I TFs, alternatively termed M-type, encode shorter proteins whose conserved regions consist solely of the MADS-box domain (Figure 1a). These TFs are predominantly implicated in embryogenesis, gametophyte development, and other reproductive processes in plants [23,36]. Some studies have indicated that the expression levels of certain Type I components change significantly under stress conditions. This alteration may play a role in the mechanisms involved in stress response [37]. Conversely, Type II TFs, known as MIKC-type, feature longer domains that encompass not only the MADS-box domain but also an additional conserved region known as the K-domain (Figure 1a). Type II transcription factors play a crucial role in plant growth, reproductive development, and the response to stress [38,39,40]. Further sub-classification by protein sequence reveals four subclasses under Type I: Mα, Mβ, Mγ, and Mδ, while Type II is divided into two subfamilies, designated as MIKC^c^ and MIKC* [41].

### 2.2. Origin and Evolution

The MADS-box gene family, which has ancient origins, has been identified in both lower and higher plants. Its lineage can be traced back to gene duplication events that occurred before the divergence of plants and animals, with the earliest common ancestor of these genes estimated to have evolved approximately 650 million years ago, prior to the Cambrian explosion [42]. Subsequently, MADS-box genes disseminated into early bryophytes, ferns, and angiosperms (Figure 1d) [43]. The cloning of the first MADS-box gene, *AG*, in *Arabidopsis* revealed its role in the regulation of floral organ development, characterized by a conserved DNA-binding domain [44]. Throughout evolutionary history, MADS-box genes have experienced multiple rounds of gene duplication and functional diversification. Research indicates that various lineages of MADS-box genes have diversified incrementally, closely linked to morphological evolution in plants [45]. For example, the ABC model proposes that A, B, and C class genes interact to regulate the development of the four distinct parts of flowers [46]. In *Arabidopsis*, the adaptive evolution of MADS-box genes primarily occurred through positive Darwinian selection (PDS). Studies have also shown that several amino acid residues within the MADS and K-domains became fixed following specific gene duplication events, suggesting that these regions played significant roles in the functional diversification of MADS-box genes [42]. Furthermore, the expression patterns of MADS-box genes have also undergone changes, with some genes expressed in gametophytic structures, such as pollen, being regarded as remnants of ancestral functions [47].

## 3. Functions of Plant MADS-Box TFs

### 3.1. Plant Growth and Reproductive Development

Plant growth and development represent a complex yet orderly series of processes that unfold under the influence of environmental cues. Seeds germinate under suitable conditions, initiating root and shoot elongation [48]. Subsequently, plants establish root systems, stems, and leaves, engaging in photosynthesis and nutrient uptake, which promote continuous growth [49]. As they transition into the reproductive phase, plants form inflorescences and flowers, undergoing sexual reproduction that culminates in fruit production [50]. Eventually, plants enter senescence, characterized by leaf abscission and eventual demise [51]. Throughout this continuum, a myriad of transcription factors orchestrates developmental transitions, among which the MADS-box gene family stands out as a critical regulator.

MADS-box TF genes play pivotal roles in root and leaf development, as well as in controlling flowering time, floral organ identity, and ovule and fruit maturation (Figure 2; Table 1) [52,53,54]. In root development, MADS-box genes exhibit multifaceted functionality. The *Arabidopsis* MADS-box gene *AGL21* promotes root elongation, while *SOC1* and *AGL24* display redundant effects as inhibitors of root growth [6,55]. In rice, *OsMADS25* acts as a positive regulator, collaborating with *OsNAR2.1* to respond to nitrate signals and regulate root system architecture [56,57,58]. In tomato, MADS-box gene *SlMADS83* enhances adventitious rooting by elevating auxin levels at the stem base [59]. In leaf development, MADS-box genes reveal complex regulatory networks and multidimensional impact pathways. The *Arabidopsis* protein APETALA3 (AP3), which can transform leaves into petal-like organs, forms ternary complexes either with *PISTILLATA* (*PI*) and *APETALA1* (*AP1*), or *PI* and *SEPALLATA3* (*SEP3*), thereby enhancing its own expression [60]. In rice, *OsMADS57* binds to the *CArG-box* in the promoter region of *D14*, directly repressing *D14* expression. It also interacts with *OsTB1*, reducing its suppression of *D14* transcription, thus regulating leaf development [61]. In tomato, overexpression of *SlMBP21* leads to imbalanced cell growth, altered expression of polarity genes, and changes in leaf structure and morphology, resulting in curled leaves [62]. During floral organ formation, the MADS-box gene family exhibits diverse functions and cooperative interactions. In *Arabidopsis*, MADS-box genes *AGL80* and *AGL61* function similarly and may work together as heterodimers in central cells, directly regulating downstream genes such as *DMETER* (*DME*) and *DD46* [63,64]. In rice, loss-of-function mutants of *OsMADS2* lead to earlier flowering, increased floral organ numbers, abnormal cell wall metabolism, and pollen sac malformation. Additionally, double interference of *OsMADS4* in an *OsMADS2* mutant background exacerbates these defects, indicating functional redundancy between *OsMADS4* and *OsMADS2* [65]. Overexpression of the tomato MADS-box gene *SlMBP22* affects multiple floral identity genes and cell expansion-related genes, resulting in significant changes in flower morphology, including alterations in organ size and color [66]. Interestingly, MADS-box genes also play a critical role in determining flowering time, reflecting intricate mechanisms that fine-tune plant adaptability. For instance, the *Arabidopsis AGL6* functions as a transcriptional repressor that directly controls flowering through the AGL6-ELF3-FT pathway [7]. In rice, *OsMADS50* activates flowering-associated genes such as *OsMADS14*, while *OsMADS51* acts as a flowering activator, relaying the signal from OsGI to Ehd1 under short-day conditions to regulate flowering time [67]. Research revealed that overexpression of *OsMADS15* results in early internode elongation, crown root development, dwarfism, and early flowering, illustrating both direct and indirect regulation of flowering time in rice [68]. In tomato, *FUL2* and *MBP20* promote flowering by inducing inflorescence initiation and synergistically suppressing inflorescence branching [69]. Furthermore, MADS-box genes play diverse roles in fruit development and maturation, illustrating how plants optimize the quality and yield of reproductive products through complex regulatory networks [67,68,69,70]. In addition, the sugarcane MADS-box genes *SOC1* and *SVP* have higher expression levels during the maturity stage of the stem, suggesting that they may be involved in the regulation of the stem maturation process [71].

**Table 1 ijms-25-13278-t001:** List of MADS-box TFs associated with growth and reproductive development.

Plant	Gene	Regulatory Site	Function	Reference
*Arabidopsis thaliana*	*AGL21*	Root	Positively regulating auxin accumulation in lateral roots	[55]
	*SOC1*	Root	Inhibiting root growth	[6]
	*AGL24*	Root	Inhibiting root growth	[6]
	*AP3*	Leaf	Promoting the transformation of leaves into petal-like organs	[60]
	*AGL80*	Flower	Promoting the expression of *DEMETER* and *DD46*	[63]
	*AGL61*	Flower	Form a heterodimer with AGL80 to promote the expression of *DEMETER* and *DD46*	[64]
	*AGL6*	Flower	Directly inhibiting the expression of the *ELF3*	[54]
	*AGL8*	Fruit	Promoting silique development	[11]
*Oryza sativa*	*OsMADS25*	Root	Activating the expression of *OsMADS27* and *OsARF7*	[72]
	*OsMADS57*	Leaf	Inhibiting the expression of *D14*	[61]
	*OsMADS2*	Flower	Promoting the expression of *PME24*, *GH9B16*, *TDR*, and *SPL*	[65]
	*OsMADS4*	Flower	Possibly functioning redundantly with *OsMADS2*	[65]
	*OsMADS50*	Flower	Activating the transcription of *OsMADS14*	[67]
	*OsMADS14*	Flower	Overexpression leading to early flowering phenotype	[73]
	*OsMADS51*	Flower	Transmitting flowering signals	[67]
	*OsMADS15*	Flower	Overexpression leading to early flowering phenotype	[68]
	*OsMADS1*	Fruit	Promoting the expression of the *OsMADS17* gene	[74]
	*OsMADS17*	Fruit	Promoting the expression of the *OsAP2-39* gene	[74]
* Solanum lycopersicum *	*SlMADS83*	Root	Inhibiting auxin synthesis	[59]
	*SlMBP21*	Leaf	Promoting the expression of *KNOX1*, *KNOX2*, *PHAN*, and *LANCEOLATE*	[62]
	*SlMBP22*	Flower	Forming dimers with *MACROCALYX (MC)*, *TM5*, and *TM29*	[66]
	*FUL1*	Flower	Synergistically inhibiting inflorescence branching with *FUL2* and *MBP20*	[69]
* Solanum lycopersicum *	*FUL2*	Flower	Inducing floral bud formation and inhibiting inflorescence branching	[69]
	*MBP20*	Flower	Inducing floral bud formation and inhibiting inflorescence branching	[69]
	*SlMADS1*	Fruit	Inhibiting the expression of ethylene biosynthesis genes	[75]
*Saccharum spontaneum*	*SOC1*	Stem	Possibly participating in hormone signal transduction pathways	[71]
	*SVP*	Stem	Possibly participating in hormone signal transduction pathways	[71]

### 3.2. Abiotic Stress

Plants activate a suite of intricate regulatory pathways to cope with adverse environmental conditions when confronted with abiotic stress such as drought, heat, and salinity, [76,77,78,79]. MADS-box proteins contribute to stress tolerance by modulating osmo-protectant gene expression, participating in hormone signaling cascades, regulating defense-related gene activity, and engaging in signal perception and transmembrane transmission (Figure 3; Table 2) [56,80,81,82,83]. These findings provide new insights into the role of MADS-box proteins in plant responses to non-stressful conditions, thereby expanding our understanding of how these transcription factors integrate stress signals into developmental programs.

**Drought Stress:** In *Arabidopsis*, the MADS-box gene *AGL16* exhibits preferential expression in guard cells and is downregulated under drought stress. It binds to the *CArG-box* present in the promoters of the *CYP707A3*, *AAO3*, and *SDD1* genes, thereby regulating their transcription and consequently affecting stomatal density and abscisic acid (ABA) levels. This suggests that *AGL16* exerts a negative regulatory effect on drought tolerance by influencing both stomatal density and ABA synthesis [84]. In rice, *OsMADS23* functions as a positive regulator, physically interacting with the SnRK2-type protein kinase SAPK9, which phosphorylates *OsMADS23* to enhance its stability and transcriptional activity. In response to osmotic stress, following activation through the ABA signaling cascade, *OsMADS23* directly stimulates the transcription of key genes involved in ABA and proline biosynthesis, such as *OsNCED2* and *OsP5CR*. Overexpression of *OsMADS23* significantly enhances drought and salt tolerance in rice, whereas knockout mutants exhibit markedly reduced resilience to these stresses [85]. In citrus, *PtrANR1*, a MADS-box TF, enhances indole acetic acid (IAA) content and promotes root growth by binding to the *CArG-box* in the *PtrAUX1* promoter, thereby improving drought stress tolerance [86]. In foxtail millet (*Setaria italica*), the expression of the MADS-box gene *SiMADS51* is critical for the drought stress response [12]. Genome-wide analysis of the MADS-box gene family in common bean (*Phaseolus vulgaris*) has elucidated the role of MADS-box proteins in drought resistance [87]. In alfalfa (*Medicago sativa*) and dragon fruit (*Selenicereus undatus*), MADS-box genes are essential for root and seed development, actively participating in responses to drought stress [37,88]. Similarly, in sheepgrass (*Festuca ovina*), MADS-box plays a crucial role in the drought stress response, promoting root growth and enhancing drought tolerance [89]. Furthermore, MADS-box genes in wheat (*Triticum aestivum*) and chili peppers also contribute to responses against cold, salt, and drought stresses [90,91].

**Salt Stress:** The roles of MADS-box proteins in plant responses to salt stress are gradually being elucidated. In *Arabidopsis*, *AGL16* has been identified as a negative regulator in the salt stress response, as it suppresses the synthesis of heat shock proteins (HSPs) and ABA signaling pathways by downregulating salt stress-related genes [92]. In rice, *OsMADS25* contributes to salt and oxidative stress tolerance through ABA-mediated regulatory pathways and ROS scavenging mechanisms [72,93]. Studies have also indicated that *OsMADS23* and *OsMADS57* exert positive regulation under drought stress [85,94]. Genomic analysis of the barley (*Hordeum Vulgare*) MADS-box gene family has revealed several members potentially implicated in salt stress responses. Among these, *HvMADS13* shows substantially increased expression following prolonged exposure to salt stress [95]. In cotton (*Gossypium hirsutum*), the MADS-box gene *GhFYF* exhibits upregulated expression under salt stress conditions. Transgenic *Arabidopsis* expressing *GhFYF* demonstrated improved seed germination and growth under varying salt concentrations, along with regulated proline content, implicating *GhFYF* in enhancing salt stress resistance [96]. Members of the MADS-box gene family in alfalfa and sheepgrass similarly contribute to the regulation of responses to salt stress and other abiotic stresses [37,89]. This reinforces the notion that MADS-box proteins play a crucial role in mediating plant adaptability under saline conditions.

**Temperature Stress:** In recent years, a series of studies have unveiled the roles of MADS-box proteins in plant responses to temperature stress. In rice, the MADS-box protein OsMADS57 interacts with *OsTB1* to directly target the defense gene *OsWRKY94* and the organogenesis gene *D14*, promoting their expression under cold stress while inhibiting their activity at normal temperatures, thereby balancing the control over cold tolerance [61]. In pepper, the MADS-box gene *CaMADS* is upregulated under cold stress. Silencing of *CaMADS* leads to increased electrolyte leakage, elevated malondialdehyde (MDA) levels, and decreased chlorophyll content in plants subjected to cold stress. Conversely, overexpression of *CaMADS* in treated plants results in significantly higher survival rates and lower levels of H_2_O_2_ and superoxide radicals under cold treatment. These findings indicate that *CaMADS* functions as a positive stress-response transcription factor, playing a role in the cold stress signaling pathway [91]. Recent studies have revealed seasonal-specific expression variations of MADS-box genes in the tea tree (*Camellia sinensis*), with several MADS-box genes exhibiting heightened expression during the winter months compared to other seasons, such as *CsFUL1a* and *CsFLC2* [97]. In kiwifruit, an FLC-like MADS-box gene, *AcFLCL*, has been found to be upregulated under cold stimulation. Its expression is regulated by histone modifications, particularly the trimethylation of lysine 4 on histone H3 (H3K4me3), which increases during cold responses, indicating multi-level epigenetic regulation. The deletion of *AcFLCL* significantly delays bud dormancy release and growth initiation in kiwifruit, suggesting that *AcFLCL* could enhance cold stress resistance by stimulating growth [34]. The MADS-box TF genes in alfalfa have been shown to exhibit upregulated expression under cold stress conditions [86], further supporting their role in cold acclimation. In apple (*Malus domestica*), silencing of the MADS-box gene *MdDAM1* results in defects in the cessation of autumn growth, providing insights that it could facilitate the cultivation of low-chill apple cultivars that are resilient to climate change [98].

**Table 2 ijms-25-13278-t002:** List of MADS-box TFs associated with stress.

Gene	Stress	Plant	Function	Reference
*AGL16*	Drought, salt	*Arabidopsis thaliana*	Inhibiting the expression of genes such as *HKT1;1*, *HsfA6a*, and *MYB102*	[92]
*OsMADS23*	Drought, salt	*Oryza sativa*	Activating the transcription of *OsNCEDs* and *OsP5CR*	[85]
*PtrANR1*	Drought	*Citrus reticulata*	Regulating positively genes related to ABA biosynthesis and ROS scavenging	[86]
*SiMADS51*	Drought	*Setaria italica*	Inhibiting the expression of genes related to plant drought resistance	[12]
*OsMADS25*	Salt	*Oryza sativa*	Enhancing salt tolerance through ABA pathways and ROS scavenging	[72]
*OsMADS57*	Salt, cold	*Oryza sativa*	Enhancing salt tolerance by activating the antioxidant system	[94]
*HvMADS13*	Salt	*Hordeum vulgare*	Significantly upregulated under salt stress	[95]
*GhFYF*	Salt	*Gossypium hirsutum*	Possibly modulating plant salt tolerance	[96]
*CaMADS*	Cold	*Capsicum annuum*	Inhibiting the generation of H_2_O_2_ and superoxide radicals	[91]
*CsFUL1a, CsFLC2*	Cold	*Camellia sinensis*	Significantly upregulated during the cold acclimation phase	[97]
*AcFLCL*	Cold	*Actinidia chinensis*	Enhancing expression during cold treatment	[34]
*MdDAM1*	Cold	*Malus domestica*	The silence of *MdDAM1* leading to the release of dormancy and bud sprouting	[98]
*NbMADS1*	*Phytophthora nicotianae*	*Nicotiana benthamiana*	Mediating processes such as stomatal closure, hypersensitive cell death, and the expression of defense-related genes through the H_2_O_2_-NO pathway	[99]
*OsMADS26*	*Magnaporthe oryzae Xanthomonas oryzae*	*Oryza sativa*	Downregulating the expression enhancing the resistance to pathogens	[83]
*GmCAL*	*Soybean Mosaic Virus*	*Glycine max*	Overexpression enhancing the SMV resistance	[100]
*TaMADS19 TaMADS117*	*Septoria tritici Blumeria graminis*	*Triticum aestivum*	The expression level was significantly upregulated after pathogen infection	[101]
*ScAGL7*	*Sporisorium scitamineum*	*Saccharum spontaneum*	Significantly upregulated after *S. scitamineum* infection	[102]

### 3.3. Biotic Stress

The response of plants to biotic stresses involves a complex network of interactions that encompasses multiple processes. Small RNAs have emerged as crucial players in plant biotic stress responses, participating in immune reactions and presenting new opportunities for plant protection [103]. Plant hormones, particularly melatonin, alleviate biotic stress by inducing defensive responses and restraining pathogen proliferation [104,105]. Additionally, fluctuations in light conditions affect plant responses to biotic stresses, influencing both physiological functions and resistance capabilities [106]. Beyond their roles in abiotic stress responses, MADS-box genes have also been implicated in biotic stress reactions. In tobacco, silencing of *NbMADS1* diminishes plant resistance against the pathogen *Phytophthora nicotianae*, indicating its positive regulatory role in systemic resistance [99]. In rice, plants with downregulated *OsMADS26* exhibit enhanced resistance to two major pathogens, *Magnaporthe oryzae* and *Xanthomonas oryzae*, suggesting that *OsMADS26* negatively regulates defense against these pathogens. Conversely, overexpression of *OsMADS26* results in a moderately increased susceptibility to *M. oryzae*, confirming its role in negatively controlling disease resistance [83]. In soybean (*Glycine max*), the *GmCAL* gene, which is induced following infection by *Soybean mosaic virus* (SMV), enhances resistance to SMV when overexpressed [100]. In bread wheat, significant changes in the expression levels of several MADS-box genes occur in response to pathogen infection. For instance, *TaMADS19* exhibits a 7- to 10-fold increase in expression level following infection by *Septoria tritici*, while *TaMADS117* shows a 3- to 7-fold downregulation after *Powdery mildew* infection [101]. In sugarcane, infection by *Sporisorium scitamineum* induces changes in floral structures, accompanied by shifts in MADS-box gene expression, indicating their involvement in flowering triggered by *S. scitamineum*. This information is crucial for formulating strategies to manage sugarcane smut [102].

## 4. Applications of Plant MADS-Box TFs in Crop Breeding

As MADS-box genes play a critical role in various essential stages of plant growth and development, adjusting their expression may increase biomass and yields, presenting significant potential for applications in food crop breeding. Furthermore, MADS-box genes are involved in plant responses to abiotic and biotic stresses, and modifying these genes could enhance the development of stress-resistant crop varieties. For example, the ectopic expression of rice MADS-box genes alters panicle emergence timing and height, demonstrating that variations in heading periods (whether early or delayed) affect the accumulation of photosynthetic products, thereby influencing grain filling during the ripening phase and ultimately impacting rice yield and quality [73]. This underscores the potential of modulating MADS-box genes to enhance agronomic traits in rice and other cereal crops [107]. In apple tree, extended exposure to cold temperatures during winter is necessary for the ability to flower and grow in the subsequent spring, a process closely associated with the MADS-box gene *MdDAM1*. Silencing this gene leads to premature breaking of dormancy and sprouting in spring, indicating that regulating the expression of related MADS-box genes could enable fruit crops to undergo regular growth and reproductive development under unfavorable environmental conditions [98]. In tomato, silencing the *SlMBP3* gene results in smaller fruits, prevents the softening of fruit flesh, and induces defects in seed development. Anatomical studies suggest that the suppression of *SlMBP3* may inhibit fruit growth by regulating placenta cell division, thereby illustrating the critical role of MADS-box genes in fruit and seed development. By manipulating the expression of MADS-box genes, we can influence the size, shape, and flavor of fruits [108]. These examples clearly prove that the MADS-box TF gene family presents significant prospects for applications in crop breeding.

## 5. Discussion

This article reviews the structure, classification, origin, and evolution of the MADS-box TFs in plants, as well as their role in plant growth, development, and stress response. As one of the largest transcription factor families in plants, the MADS-box gene family has been extensively studied for its contributions to plant growth and development; however, its involvement in plant stress response has been examined to a lesser extent. Especially, the mechanisms of action in tropical plants remain a relatively underexplored area. Sugarcane, serving as the primary source of sugar, is a vital tropical cash crop. Its essential contributions to the production of sugarcane and ethanol fuel, as well as various industrial raw materials underscore its considerable impact on the agricultural economy and sustainable development in numerous countries [17,18,19]. Consequently, investigating the mechanisms of MADS-box TFs in sugarcane is crucial for the sugar industry.

MADS-box TFs regulate the formation of floral organs and contribute to the architecture of roots, stems, and leaves. Moreover, they can even interact with each other to form complex regulatory networks that ensure proper plant development at various stages of the life cycle. Based on existing research findings, we can reasonably hypothesize about the potential impacts of MADS-box genes on the growth and reproductive development of sugarcane. Given that *AGL21*, *SOC1*, *AGL24*, along with *OsMADS25* and *SlMADS83* have been shown to shape root structure and function in other crops [6,55,58,59], analogous genes in sugarcane may finely tune nitrogen sensing and auxin balance to optimize the root network for enhanced water and nutrient absorption efficiency. During stem maturation in sugarcane, the high expression levels of MADS-box genes such as *SOC1* and *SVP* suggest their involvement in regulating stem development [71]. Inspired by *AP3*, *OsMADS57*, and *SlMBP21* [60,61,62], the counterpart MADS-box genes may refine leaf geometry and physiological performance in sugarcane, thereby boosting photosynthetic efficiency and stress resilience. Echoing the effectiveness of *AGL80*, *AGL61*, *OsMADS2*, *OsMADS4*, and *FUL1* in flower organ development and flowering synchronization [64,65,70], sugarcane likely possesses similar gene function systems that control flower quantity and spatial–temporal distribution, ensuring reproductive success. Mechanisms involving *AGL6*, *OsMADS50*, *OsMADS51*, *OsMADS15*, *FUL2*, and *MBP20* across multiple crops reveal refined pathways in sugarcane that aim for optimal resource allocation and reproductive strategy optimization [7,8,67,68,109]. Under the guidance of *AGL8*, *OsMADS1*, *OsMADS17*, and *SlMADS1* in fruit development and maturation [11,74,75], those latent genes may promote sugar accumulation, seed setting rate, and commercial value, catering to modern sugarcane industry.

In response to both abiotic and biotic stressors, MADS-box TFs exhibit considerable regulatory capabilities, such as drought, salinity, and temperature fluctuations, as well as bacteria, fungi, and viruses. Drawing upon current research findings, we can reasonably propose that MADS-box genes may have a significant role in mediating the responses of the important tropical crop, sugarcane, to various environmental stressors. Akin to those in *Arabidopsis*, MADS-box genes are anticipated to play a central role in regulating drought responses, particularly through the modulation of stomatal behavior, ABA synthesis, and biochemical pathways that involve proline accumulation and the production of other osmoregulatory molecules in sugarcane. Homologous genes in sugarcane, such as *AGL16* in *Arabidopsis* and *OsMADS23* in rice [84,85], may contribute to stomatal dynamics, ABA signaling, and water management, thereby aiding plants in maintaining survival functions under water-deficit conditions to cope with drought stress. In rice and other species, MADS-box genes, including *OsMADS25* and *OsMADS23* [85,93], delineate a clear response pathway to salt stress, suggesting that their analogous genes may also play vital roles in sustaining salt homeostasis, regulating osmotic pressure, and implementing antioxidant mechanisms in sugarcane. Furthermore, drawing on phenomena observed in tea, kiwi, and other species [34,97], MADS-box genes in sugarcane may be implicated in regulating cold stress responses by influencing growth rhythms and dormancy processes, thereby assisting in the maintenance of physiological homeostasis in high-temperature environments. Notably, *OsMADS26* negatively regulates resistance to *M. oryzae* and *X. oryzae* in rice [83], implying that the corresponding MADS-box members may involve in defense against fungal diseases in sugarcane. Investigating how sugarcane MADS-box TFs are regulated in relation to pathogen recognition, immune signaling, and the expression of disease resistance genes could provide valuable insights into plant-pathogen interactions. Considering the role of *GmCAL* in combating SMV in soybean [100], it is plausible that MADS-box genes in sugarcane may also contribute to resistance against bacterial and insect attacks, particularly at critical nodes involved in regulating host–pathogen interactions. Concerning the observed expression changes of *TaMADS19* and *TaMADS117* following infections by wheat *S. tritici* and *P. mildew* [101], as well as the expression alterations of MADS-box genes after sugarcane smut infection [102], it can be inferred that analogous MADS-box TFs in sugarcane may possess the capability to confer broad-spectrum resistance to various diseases. Therefore, evaluating the functionality of sugarcane MADS-box genes under the pressure of multiple pathogens is essential for the development of multi-resistant crop varieties.

Taken together, MADS-box TFs may play a significant role in the growth, development, and environmental adaptation of sugarcane. A thorough exploration and decoding of sugarcane’s unique MADS-box genes are likely to open a new chapter in the creation of high-quality and efficient sugarcane seeds, which holds substantial value for both scientific research and industry.

## 6. Prospects and Perspectives

The MADS-box family is integral to plant growth, development, and stress response. As molecular biology and genetic technologies advance, our understanding on the MADS-box gene family will deepen, further enhancing its potential applications in crop improvement. Considering the established role of MADS-box genes in other plant species, it is highly probable that they play a crucial role in regulating the growth cycle of sugarcane and enhancing its adaptability to environmental conditions. Previous studies have demonstrated that MADS-box genes are essential for sugarcane stem development and stress response [71,102]. Given the complex genomic structure and polyploid nature of sugarcane, the mining and functional characterization of MADS-box gene family presents numerous challenges. Therefore, it is essential to investigate the versatile functions of the MADS-box gene family in sugarcane. With the rapid advancements in sequencing technology and bioinformatics tools, an increasing number of MADS-box genes have been identified, establishing a solid foundation for further exploration of their functions. Future research trends may encompass but are not limited to the following directions:Investigation of the homology of MADS-box genes between sugarcane and other tropical crops, as well as the common and specific expression patterns across different species, to establish a foundation for cross-species functional predictions;Identification and analysis of the expression patterns of sugarcane MADS-box genes in response to specific environmental stimuli, elucidating their modes of action and mechanisms, with the aim of discovering candidate genes that can be utilized to breed new varieties with enhanced stress tolerance and improved sugar production efficiency;Integration of epigenetic and proteomic data to comprehensively evaluate the dynamic regulatory network of MADS-box genes during sugarcane growth and environmental adaptation, thereby offering multidimensional support for understanding gene functions and designing effective breeding strategies;Employing modern gene editing technologies, such as CRISPR-Cas9, to functionally verify the sugarcane MADS-box genes, thereby achieving precise control over their regulatory networks.

## Figures and Tables

**Figure 1 ijms-25-13278-f001:**
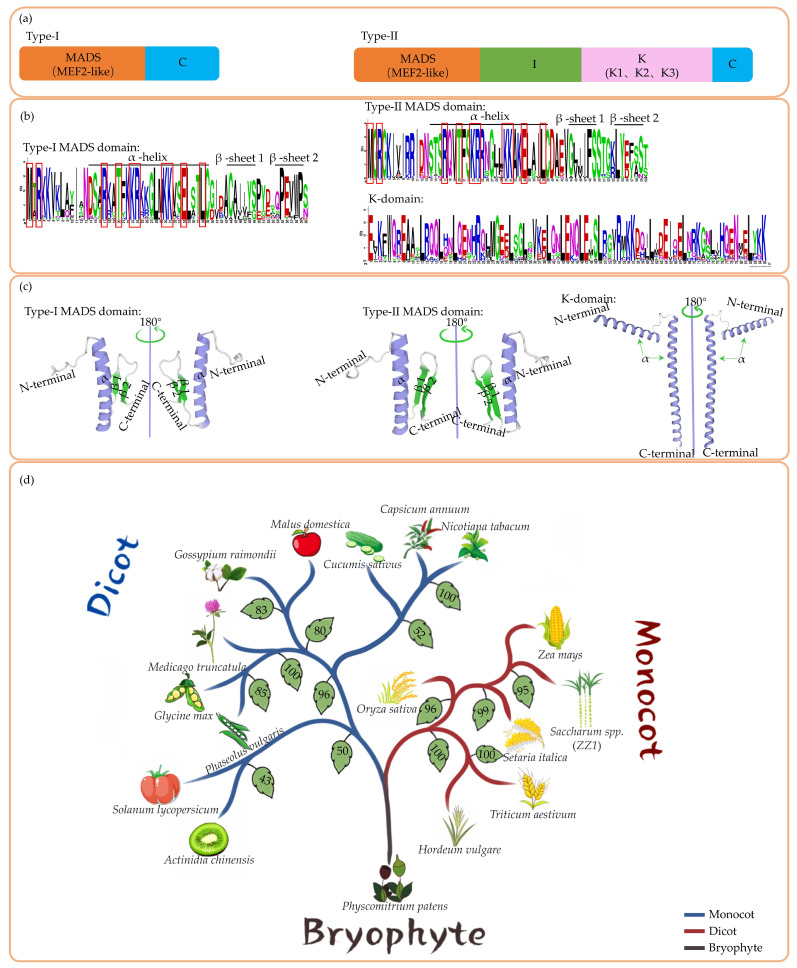
The structure and evolution of MADS-box proteins. (**a**) Structural differences between Type I and Type II MADS-box proteins; (**b**) Protein sequences of the MADS-box domains in Type I and Type II MADS-box TFs, along with the K-domains found in Type II MADS-box TFs. The red box highlights amino acids that are completely conserved across all MADS-box domains; (**c**) Three-dimensional protein structure diagrams of the Type I and Type II MADS-box domains, as well as the K-domain; (**d**) Distribution of MADS-box family genes in 17 Plantae species. The phylogenetic tree was constructed based on NCBI taxonomy browser. https://www.ncbi.nlm.nih.gov/Taxonomy/CommonTree/wwwcmt.cgi (accessed on 20 October 2024) and MEGA version 11, with *Physcomitrium patens* serving as an outgroup. The blue sections represent dicotyledonous plants, the orange sections represent monocotyledonous plants, and the green sections indicate the confidence levels of the evolutionary branches.

**Figure 2 ijms-25-13278-f002:**
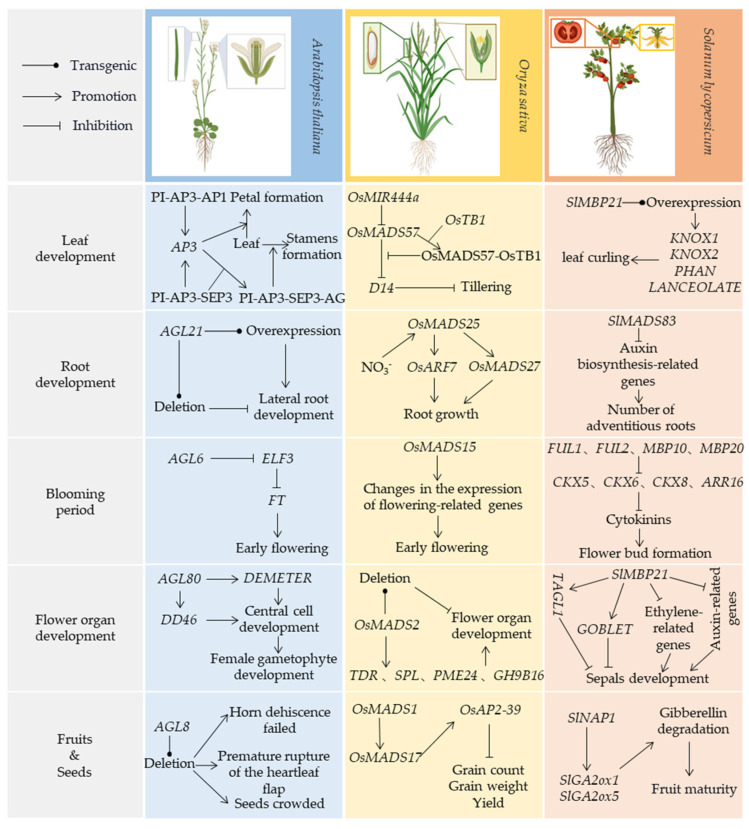
Regulatory mechanisms of MADS-box genes in various plant species during developmental processes.

**Figure 3 ijms-25-13278-f003:**
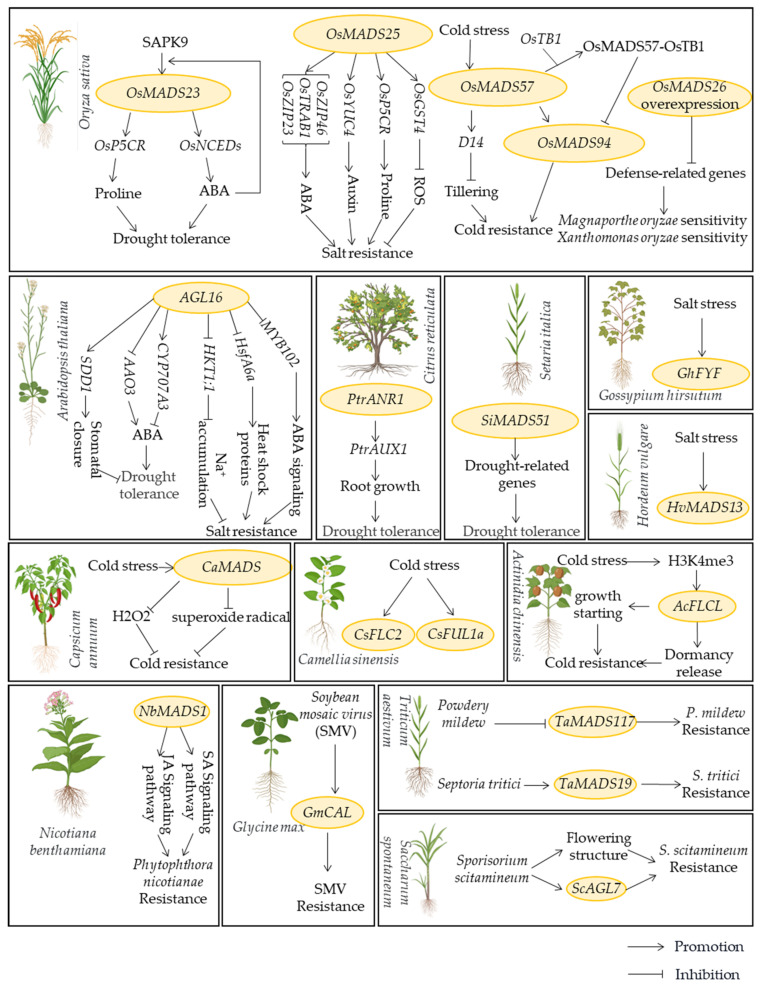
MADS-box genes are involved in abiotic stress response in different developmental processes in several plants.

## Data Availability

Not applicable.

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
