# Peer review of "Evolution and Function of MADS-Box Transcription Factors in Plants"

_ijms, 2024, doi:10.3390/ijms252413278_

Round 1
Reviewer 1 Report
Comments and Suggestions for Authors
The analysis of this paper is comprehensive, but there are still many small problems that need to be modified.
1. Keywords are too long, shorten them as much as possible, and choose some key information as keywords.
2. The introduction part of the introduction is too general, and a comprehensive and detailed introduction is needed to introduce the studies on the regulation of yield or stress resistance of some key genes of MADS-box family.
Author Response
Dear reviewer,
We are glad to receive your valuable comments and suggestions to our manuscript. Thank you for your kind consideration on this manuscript “Evolution and Function of MADS-box Transcription Factors in plants” (Manuscript ID: ijms-3353861). Without your professional reviews, this manuscript would not be as smooth and more persuasive as what it is now. Thank you very much!
We have amended the manuscript according to all the opinions, suggestions, and comments of the reviewers and all the changes have been marked up in the text by red font. The responses to all the comments and suggestions are itemized as follows:
Comment 1: The analysis of this paper is comprehensive, but there are still many small problems that need to be modified.
Response 1: Thank you for your insightful comments and the opportunity to enhance the clarity and impact of our manuscript. This is to confirm that all the comments and suggestions raised by you and the other anonymous reviewer have been incorporated into the revised manuscript. We appreciate the time and effort you have devoted to review our work. Thanks again.
Comment 2: Keywords are too long, shorten them as much as possible, and choose some key information as keywords.
Response 2: Thanks to your advised comments. We have revised the “Keywords” to the following: “MADS-box; Evolution and Function; Stress Response; Crop Improvement; Perspectives”.
Comment 3: The introduction part of the introduction is too general, and a comprehensive and detailed introduction is needed to introduce the studies on the regulation of yield or stress resistance of some key genes of MADS-box family.
Response 3: Thank you for your valuable comments. Based on your suggestions, we have revised the “Introduction” section to offer a thorough and detailed overview of the research concerning plant growth and development, stress resistance, and the breeding of key genes within the MADS-box family. The revised content can be found in lines 45 to 73.
Details are as follows:“The modification of MADS-box genes involved in flower organ development can significantly alter traits such as petal number, color, and flowering time, making them particularly valuable in ornamental plant breeding [5]. For instance, overexpression of AGL24 has been shown to promote flowering and facilitate the conversion of flower meristems into inflorescence meristems. Moreover, there is an interaction between SOC1 and AGL24 in the regulatory network governing flowering time [6,7]. Additionally, AGL6 acts as a transcriptional suppressor, promoting the transition to flowering by directly inhibiting ELF3 [6,7]. In tomatoes (Solanum lycopersicum),FUL1 and FUL2 exhibit redundant functions and are implicated in various aspects of the fruit ripening process, including cell wall modification, the production of cuticle components and volatile substances, as well as the accumulation of glutamic acid [8]. It is thus deduced that the regulation of MADS-box genes can influence the size, shape, flavor, and other characteristics of the fruit, thereby demonstrating their utility in fruit cultivation [8]. Furthermore, MADS-box TFs are associated with plant responses to stress, and manipulating these genes has the potential to develop crop varieties that exhibit enhanced stress resistance [9,10]. Silencing of CaAGL8 significantly reduces both heat resistance and the sensitivity of pepper (Capsicum annuum) to bacterial infections under conditions of room temperature, as well as elevated temperature and humidity. In transgenic Arabidopsis, overexpression of DgMADS114 and DgMADS115 has been shown to enhance the tolerance to polyethylene glycol (PEG), sodium chloride (NaCl), abscisic acid (ABA), and heat stress [11–13]. Regarding the involvement of MADS-box TF genes in various stages of plant growth and development, the expression modifications to these genes hold significant promise for enhancing biomass and yield, which could be crucial for plant breeding. For example, AGL23 plays a role in the early phases of gametogenesis, AGL80 influences central cell differentiation, and AGL62 inhibits cell differentiation while promoting nuclear proliferation during the initial stages of endosperm development [14]. Interestingly, NlMADS4, NlMADS30, and NlMADS69 are specifically and highly expressed during the development of rambutan (Nephelium lappaceum) peel, indicating their key roles in this process [15].”
Again, we want to confirm that we have amended the manuscript according to all your suggestions. Thanks again for your quick processing and professional editing of this manuscript.
Any questions, we will be more than happy to answer. Looking forward to hearing from you soon.
Sincerely yours,
Youxiong Que and Qibin Wu
2024-12-07
Reviewer 2 Report
Comments and Suggestions for Authors
The review paper is focused on interesting topic regarding the evolution and function of MADS-box transcription factors in plants. The MADS-box transcription factor (TF) gene family is pivotal in various aspects of plant biology, particularly in growth, development, and environmental adaptation. It comprises Type I and Type II categories, with the MIKC-type subgroups playing a crucial role in regulating genes essential for both the vegetative and reproductive stages of plant life. The Authors particularly emphasized implications for crop enhancement, especially in light of recent advances in understanding the impact on sugarcane (Saccharum spp.), a vital tropical crop. The review was concentrated on significance of MADS-box TFs as promising targets for future research in crop science.
In my opinion, the manuscript may be accepted in the current form.
It was a pleasure to read this manuscript. The paper is very well prepared from the formal and substantive scientific side. I do not raise any objections or recommendations to make corrections. I only suggest checking the minor style and grammar use of the English language.
Author Response
Dear reviewer,
We are glad to receive your valuable comments and suggestions to our manuscript. Thank you for your kind consideration on this manuscript “Evolution and Function of MADS-box Transcription Factors in plants” (Manuscript ID: ijms-3353861). Without your professional reviews, this manuscript would not be as smooth and more persuasive as what it is now. Thank you very much!
We have amended the manuscript according to all the opinions, suggestions, and comments of the reviewers and all the changes have been marked up in the text by red font. The responses to all the comments and suggestions are itemized as follows:
Comment 1: The Authors particularly emphasized implications for crop enhancement, especially in light of recent advances in understanding the impact on sugarcane (Saccharum spp.), a vital tropical crop. The review was concentrated on significance of MADS-box TFs as promising targets for future research in crop science.
In my opinion, the manuscript may be accepted in the current form.
It was a pleasure to read this manuscript. The paper is very well prepared from the formal and substantive scientific side. I do not raise any objections or recommendations to make corrections. I only suggest checking the minor style and grammar use of the English language.
Response 1: Thank you for your feedback. Based on your comments, I have meticulously proofread the minor stylistic and grammatical aspects of the English language. We also thanks for your high affirmation and good confirmation.
Again, we want to confirm that we have amended the manuscript according to all your suggestions. Thanks again for your quick processing and professional editing of this manuscript.
Any questions, we will be more than happy to answer. Looking forward to hearing from you soon.
Sincerely yours,
Youxiong Que and Qibin Wu
2024-12-07